# Revealing the Angiogenic Signature of *FH*-Deficient Breast Cancer: Genomic Profiling and Clinical Implications

**DOI:** 10.3390/cancers17182942

**Published:** 2025-09-09

**Authors:** Liat Anabel Sinberger, Noa Keren-Khadmy, Assaf Goldberg, Tamar Peretz-Yablonski, Amir Sonnenblick, Mali Salmon-Divon

**Affiliations:** 1Department of Molecular Biology, Ariel University, Ariel 4077625, Israel; 2Institute of Oncology, Tel Aviv Sourasky Medical Center, Tel Aviv 6423906, Israel; 3Hadassah Medical Center, Faculty of Medicine, Hebrew University of Jerusalem, Jerusalem 9112001, Israel; 4Sackler Faculty of Medicine, Tel Aviv University, Tel Aviv 6997801, Israel; 5Adelson School of Medicine, Ariel University, Ariel 4077625, Israel

**Keywords:** fumarate hydratase (*FH*), breast cancer (BC), angiogenesis, anti-VEGF treatment, targeted therapy, biomarker

## Abstract

Breast cancer is a complex disease influenced not only by genes but also by the way cancer cells generate and use energy. One rare change is the loss of a gene called fumarate hydratase (*FH*), which normally helps cells produce energy efficiently. We examine how this change influences how tumors grow and respond to treatment. By studying thousands of breast cancer samples, we found that tumors missing *FH* tend to create a special environment that helps them grow and survive, especially by encouraging new blood vessel formation. We also describe a patient with this *FH* change who had a remarkable and long-lasting response to a therapy that blocks blood vessel growth (known as anti-VEGF treatment). These findings suggest that identifying this rare energy-related alteration may help clinicians determine which patients are most likely to benefit from this therapy. Such insights could contribute to advancing more personalized approaches to breast cancer care.

## 1. Introduction

Targeting tumor angiogenesis represents a key therapeutic approach in the treatment of advanced breast cancer (BC), most notably through anti-VEGF therapies [1,2]. The role of bevacizumab in the treatment of stage IV BC has been controversial due to mixed evidence regarding its efficacy. Bevacizumab was initially granted accelerated approval for use in combination with paclitaxel for first-line treatment of HER2-negative metastatic BC based on improvements in progression-free survival (PFS) observed in clinical trials [3]. However, subsequent studies failed to demonstrate a significant improvement in overall survival (OS) with bevacizumab, and safety concerns, including risks of hypertension, proteinuria, and thromboembolic events, led the FDA to revoke its approval for BC in 2011 [4]. This highlights the complexity of anti-VEGF effects in BC and the need for better predictive biomarkers to identify patients who might benefit from bevacizumab therapy.

Emerging evidence suggests that alterations in metabolic enzymes may contribute to angiogenic signaling in cancer, implicating fumarate hydratase (*FH*) as a potential contributor in this process. FH is an enzyme in the tricarboxylic acid (TCA) cycle that catalyzes the conversion of fumarate to malate. The gene responsible for encoding *FH* is located on chromosome 1q42.2. Recent research has identified *FH* as a tumor suppressor gene involved in cancer development and progression, highlighting its potential as a therapeutic target [5,6,7].

Heterozygous germline mutations in the *FH* are associated with hereditary leiomyomatosis and renal cell carcinoma (HLRCC), an autosomal dominant cancer syndrome that significantly increases the risk of developing type II papillary kidney cancer. This form of cancer is known for its aggressive growth and early metastasis [5,8,9].

Genetic alterations resulting in *FH* loss of function lead to the accumulation of fumarate, causing dysregulation of cellular metabolism and signaling pathways. In HLRCC, FH-deficient cells, defined by reduced or absent FH activity, show impaired oxidative phosphorylation and transition to aerobic glycolysis. These cells exhibit reduced levels of AMPK, p53, and DMT1, resulting in low cellular iron levels. Additionally, the elevation of fumarate stabilizes HIF-1α (hypoxia-inducible factor 1α) and increases VEGF (vascular endothelial growth factor) expression, facilitating rapid cell growth. The effects of *FH* deficiency suggest several drug development avenues, including VEGF inhibitors, DNMT (DNA methyltransferases) inhibitors, PARP inhibitors, and LDHA (lactate dehydrogenase) inhibitors [5,6,7].

In vivo studies of kidney-specific *Fh1* deletion have shown that *FH* loss induces stabilized levels of HIF-1α and NRF2, which are involved in cell proliferation and angiogenesis [10,11,12].

Advances in genetic testing have improved the ability to predict the prevalence of *FH* germline mutations in the population. Recent studies have estimated that pathogenic or likely pathogenic *FH* variants occur in 1 in 901 to 1 in 1252 individuals. Additionally, 1.05% of breast cancer (BC) patients (389 out of 36,966) have been found to carry *FH* pathogenic or likely pathogenic variants [13,14].

In this study, we present a novel treatment approach in a BC patient with *FH* mutations using VEGF inhibitors. We also investigate how *FH* alteration affects BC tumors to better understand the mechanism behind VEGF inhibitor therapy.

## 2. Materials and Methods

### 2.1. Data Resource and Study Design

We collected clinical and genomic aberration data for 10,953 cancer patients from the TCGA pan-cancer study (Appendix A) and an additional 9163 BC samples from twelve studies (Table 1) using cBioPortal for Cancer Genomics [15,16].

Normalized gene expression profiles, probe annotation, and clinical information for 1073 samples from the TCGA dataset [29,30] were downloaded using the MetaGxBreast (version 1.12.0) R package [31]. In addition, information on somatic non-synonymous mutations, structural variance (SV), copy number alterations (CNAs), putative arm-level copy number (888 samples), and protein expression z-scores measured by mass spectrometry (105 samples) was downloaded from cBioPortal for Cancer Genomics.

Data processing was conducted using R studio software (version 4.2.2) [32]. TCGA samples were categorized based on *FH* RNA and protein expression levels.

We defined wild-type (WT) samples as those without *FH* mutations or SV and with diploid *FH* status, and “*FH* aberration” as samples with a combination of CNA shallow deletion and/or 1Q arm loss, and/or samples with *FH* mutation.

All clinical information in the Case Study Section was collected and published following ethical approval (Helsinki ethics approval number TLV-19-449), with informed consent obtained from the patient for use of their anonymized data.

The study design is presented in Figure 1.

### 2.2. Calculation of Angiogenesis Score

Angiogenesis scores were calculated for each TCGA breast cancer sample using the GSVA R package [33] (version 2.2.0) based on the HALLMARK_ANGIOGENESIS gene set obtained from the MSigDB database [34,35] R package (v25.1.1).

### 2.3. Differentially Expressed Gene (DEG) Identification

Differential gene expression analysis was conducted using limma [36] (version 3.54.1) and edgeR [37] (version 3.40.2) R packages. DEGs were identified across the various groups, with statistical significance defined as a false discovery rate (FDR) ≤ 0.05 and a fold change ≥ 2.

### 2.4. Estimation of Tumor-Infiltrating Immune Cells

Tumor-infiltrating immune cell proportions in TCGA samples were assessed using data from the TIMER3 database [38,39] (https://compbio.cn/timer3/ accessed on 21 August 2025). We analyzed the infiltration estimates derived from the CIBERSORT deconvolution method [40] employing the absolute score as the quantitative metric.

### 2.5. Weighted Correlation Network Analysis (WGCNA)

Gene expression profiles from TCGA samples were subjected to signed network analysis using the WGCNA [41,42] R package (version 1.70.3). The network was built with a power (β) of 12, the default parameter for signed networks, ensuring a scale-free topology model fit of 0.8 (R^2^ > 0.8) [43] (Appendix A). Modules with high similarity were merged using a cutline of 0.25 (the recommended threshold, corresponding to eigengene correlation > 0.75 [43], Appendix A). Correlated modules were identified based on their association with *FH* mRNA and protein expression traits. For the selected modules, we calculated module membership (MM) and gene significance (GS).

### 2.6. Identifying Hub Genes

Hub genes are defined as those within candidate modules that show strong correlations with specific traits in the WGCNA analysis. Identification of hub genes was performed as previously described [44]. Briefly, unless stated otherwise, the top 10% of genes in a candidate module with module membership (MM)  >  0.8 and gene significance (GS)  >  0.2 were considered hub genes [45]. These hub genes were used as input to perform protein–protein interaction (PPI) analysis using the Search Tool for Retrieval of Interacting Genes/Proteins (STRING) [46] database. Genes with the highest connectivity in each PPI network were identified as key hub genes, as they likely play critical roles in the module [45].

### 2.7. Pathways Enrichment Analysis

Gene set enrichment analysis (GSEA) was performed through the WebGestalt 2024 [47] tool against the Hallmark pathways gene sets obtained from the MSigDB database [35,48] using the list of differentially expressed genes (DEGs) as input. Pathways with FDR  ≤  0.05 are shown.

Overrepresentation analysis (ORA) was performed using GeneAnalytics [49] and Metascape [50] using genes from WGCNA-selected modules. Pathways with FDR  ≤  0.05 are shown.

### 2.8. Statistical Analysis

Statistical analyses were performed utilizing the R 4.2.2 statistical framework. Differences in categorical variables among groups were analyzed using Fisher’s exact test. Continuous parameters among distinct groups were analyzed using the nonparametric Kruskal–Wallis test, followed by the Dunn post hoc test. Kaplan–Meier survival curves were generated and visualized using the R packages Survival [51] (Version 3.5.3) and Survminer [52] (Version 0.4.9), with *p*-values calculated by Cox regression analysis. Plots were generated using the ggplot2 (Version 3.4.1) [53] R package.

## 3. Results

### 3.1. Prevalence of FH Alterations in Cancer Patients

We analyzed *FH* aberrations across different cancer types using data from pan-cancer studies and breast cancer (BC) studies. In pan-cancer studies, the mean prevalence rates were 0.54% for *FH* mutations, 7.75% for copy number alteration (CNA) deletions (including shallow and deep deletions), and 5% for 1q arm loss (Appendix A). In BC, we examined 9163 primary and metastatic samples from twelve studies (Table 1) and found mean prevalence rates for *FH* mutations of 0.36% (95% CI: 0.29–0.43% across 13 studies), significantly below the pan-cancer prevalence mean (one-sample t-test, t(12) = −5.4, *p* < 0.001), 1.72% for CNA deletions, and 1.13% for 1q arm loss. While the estimate was consistent across studies, the rarity of *FH*-deficiency reduces statistical power and warrants cautious interpretation.

To understand how *FH* deficiency affects clinical features in BC, we classified samples based on mutation, CNA status, and 1q arm level. We then examined the clinical properties of 94 samples with *FH* aberrations compared to 9069 WT samples (Appendix A). Our analysis revealed that *FH* aberrations occurred more frequently in primary samples than in metastatic samples (Fisher’s exact test, N = 9074, odds ratio (OR) = 1.8, *p*-value < 0.05). In primary samples, these aberrations were more commonly found in tumors larger than 2 cm (Fisher’s exact test, N = 990, OR = 2.32, *p*-value < 0.05) and in older patients (Kruskal–Wallis test, N = 2301, *p* < 0.05).

Table 2 describes the mutational landscape of identified *FH* mutations classified as likely oncogenic according to the OnkoKB dataset [54] in 11 BC-mutated samples from nine patients. *FH* mutations were identified in both metastatic and primary BC samples, encompassing a range of mutation types and variant classes. The most recurrent alteration was a fusion event involving *FH*, observed in combination with PDE1C, RGS7, and MIR-1273E/1273E genes, predominantly in metastatic samples. Additionally, primary tumor samples harbored frameshift insertion (L208Vfs9), nonsense mutation (Q237*), and splice region variant (X186_splice), while metastatic samples harbored missense mutation (M336K) and frameshift deletion (P63Ifs*9), indicating potentially diverse mechanisms of *FH* inactivation. Collectively, these findings suggest that *FH* alterations in BC are rare and span a spectrum of structural and point mutations.

### 3.2. Effect of *FH* Deficiency on Primary Breast Cancer Tumors

Previous studies have shown that *FH*-deficient renal cancer cells exhibit impaired oxidative phosphorylation and increased aerobic glycolysis. This leads to dysregulated gene expression in pathways that promote tumor growth, including angiogenesis and invasion [5,55,56]. To investigate the effect of *FH* deficiency on primary BC tumors, we analyzed data from the TCGA dataset. We explored the correlation between *FH* mRNA and protein expression, finding a significant positive correlation (N = 105, R = 0.62, *p* < 2.2 × 10^−16^, Figure 2a). Given the low number of samples having known *FH* aberrations and the limited availability of protein expression data in the dataset, we classified BC samples based on their *FH* RNA expression levels. We defined the “*FH*-deficient” group as the 25% of samples with the lowest *FH* expression, with the rest defined as “high-*FH*”. When assessing the prevalence of *FH* aberrations and mutations across these groups, we observed that such aberrations were significantly more frequent in the *FH*-deficient expression group (Fisher’s exact test, N = 1059, *p* = 6.82 × 10^−14^, OR = 19.5, Figure 2b).

To account for potential confounding by clinical variables, we calculated an angiogenesis score for each sample and performed both univariate and multivariate linear regression analyses, with angiogenesis score as the dependent variable and *FH* expression, age, tumor size, and PAM50 subtype as predictors (Appendix A). The multivariate analysis demonstrated that *FH* expression retained a highly significant effect on the angiogenesis score (*p* = 1.97 × 10^−14^), independent of age, tumor size, or molecular subtype.

### 3.3. Molecular and Clinical Consequences of FH Deficiency in Breast Cancer

To investigate the impact of *FH* deficiency on molecular signaling, we conducted differential gene expression (DGE) analysis comparing *FH*-deficient and high-*FH* expression groups. We identified 24 downregulated and 54 upregulated genes (logFC = 1, FDR ≤ 0.05). Among them, SCUBE2 has been reported as upregulated in BC metastasis [57], while CA9, PRAME, and ELF5 are downregulated in BC metastasis [57,58]. CHAD has been reported to play a role in focal adhesion [59] (Figure 3a, Appendix A). These results suggest that *FH* deficiency may influence breast cancer progression by modulating genes associated with promoting metastasis and cell adhesion.

GSEA against the MSigDB Hallmark collection [35] based on the DEGs revealed the expected downregulation of oxidative phosphorylation and the TCA cycle. In contrast, angiogenesis and epithelial–mesenchymal transition (EMT) pathways were upregulated (Figure 3b), suggesting a potential benefit from targeting the VEGF pathway in breast cancer patients with *FH* aberrations.

Previous studies have demonstrated that metabolic reprogramming can reshape the tumor microenvironment (TME) and alter the composition of tumor-infiltrating immune cells [60]. To investigate this, we applied the CIBERSORT algorithm [40] to deconvolute bulk RNA expression data and estimate 22 subsets of tumor-infiltrating lymphocytes (TILs). Comparison of *FH*-deficient and high-*FH* breast tumors (Appendix A) revealed a significant increase in activated mast cells, which are implicated in tumor proliferation and angiogenesis [61], with an additional increase detected in resting memory CD4^+^ T cells (Figure 3c).

To explore the prognostic significance of *FH* deficiency, we analyzed OS and recurrence outcomes stratified by molecular subtype. Kaplan–Meier and Cox regression analyses stratified by PAM50 subtypes demonstrated that *FH*-deficient tumors were associated with significantly worse overall survival in the basal subtype (HR = 6.0, 95% CI 2.09–17.23, *p* < 0.001, Figure 3d), while no consistent associations were observed in luminal and HER2-positive subtypes. Disease-free survival did not differ significantly across groups (Appendix A).

### 3.4. Weighted Correlation Network Analysis

We applied weighted correlation network analysis (WGCNA) to all TCGA samples with *FH* expression data to identify gene modules associated with *FH* mRNA and protein expression (Figure 4a).

Our analysis identified a positive correlation of the MEdarkgrey module (correlation = 0.64, *p*-value = 2 × 10^−126^) and a negative correlation of the MEgreen module (correlation = −0.35, *p*-value = 3 × 10^−32^) to *FH* mRNA expression. Additionally, we observed a negative correlation between the MEpink module and FH protein expression (correlation = −0.24, *p*-value = 1 × 10^−15^) (Figure 4a).

Overrepresentation analysis (ORA) of the MEdarkgrey genes indicated enrichment of the TCA cycle pathway (Figure 4b). ORA of MEgreen genes with high module membership (MM > 0.7) showed enrichment in pathways related to blood vessel development, cell adhesion, and VEGF signaling (Figure 4b). Similarly, the MEpink module genes were enriched for pathways involved in blood vessel development and cell adhesion (Figure 4b).

### 3.5. Identifying Key Hub Genes

Key hub genes are those highly connected in protein–protein interaction (PPI) networks and may serve as potential biomarkers or predictive markers [45,62]. In the MEgreen module, we identified seven hub genes: CDH5, MMRN2, ADCY4, CLDN5, VWF, CD34, and PECAM1 (Figure 5a,b). These genes, which were upregulated in the *FH*-deficient group, play roles in cell adhesion, blood vessel development, and cell migration (Figure 5c,d).

In the MEpink module, we identified four key hub genes: FBN1, TIMP2, CDH11, and PDGFRA (Figure 5e). These genes are negatively correlated to FH protein expression and are involved in angiogenesis (Figure 5f,g). All together these results support VEGF inhibition as a targeted therapeutic strategy for *FH*-deficient BC tumors.

### 3.6. Case Study

To highlight the potential clinical significance of these findings, we present a case of a 71-year-old woman with a medical history of type II diabetes and hysterectomy (uterine fibroids–myoma) who presented in June 2021 with a new firm mass above her left breast. CT imaging revealed a heterogeneous mass in the chest wall, measuring 6 cm, involving the pectoralis muscle, with metastases to the lungs and regional lymph nodes. Biopsy of the breast lump confirmed invasive ductal carcinoma (IDC) and triple-negative GATA3 positive, with a Ki-67 index of 50.

Next-generation sequencing (NGS) testing revealed a tumor mutational burden (TMB) of 0 and microsatellite stability (MSS). Somatic mutations were identified in PTEN, MLL2, and TP53. In addition, an *FH* mutation (H318Y, 952C > T) was detected with a variant allele frequency of 56.2% and has been reported in the ClinVar database as a likely pathogenic or pathogenic germline mutation (submitted by an expert panel or multiple submitters). Her CPS PDL1 status was below 10, and no germline BRCA mutations were identified. It was therefore decided, with the diagnosis of stage 4—triple negative breast cancer, to treat the patient with palliative chemotherapy with paclitaxel. Based on international guidelines and the potential benefit of VEGF inhibition due to the *FH* mutation, the tumor board suggested adding the anti-VEGF bevacizumab to the protocol. She received paclitaxel 80 mg/m^2^ weekly and bevacizumab 10 mg/kg bi-weekly, with a dramatic long-lasting response (Figure 6). Due to cumulative toxicity and neuropathy, paclitaxel doses were reduced and subsequently switched to capecitabine (Xeloda) 1500 mg BID (days 1–14 of a 21-day cycle), while bevacizumab was continued with no evidence of progression. The patient remained on this regimen until June 2024, achieving a progression-free survival of two years, at which point disease progression was observed in the left breast and lungs.

## 4. Discussion

The mixed clinical outcomes associated with bevacizumab in metastatic BC highlight a broader issue in precision oncology: the need for reliable biomarkers to guide anti-angiogenic therapy. In this context, our study explores the potential role of *FH* alterations as a contributing factor to angiogenic signaling and a possible biomarker of therapeutic response. Generally, *FH* aberrations (mutations, CNA, and 1q arm level) prevalence in BC is lower than in other cancer types. Notably, in BC, *FH* aberrations were more common in primary tumor samples and appeared more frequently in older patients.

Our analysis of the TCGA dataset suggests that *FH* expression levels influence the metabolic and signaling pathways in primary BC tumors. The downregulation of oxidative phosphorylation, glycolysis, and TCA cycle pathways in low-*FH* tumors indicates a shift in energy metabolism, suggesting a distinct metabolic reprogramming characteristic of these tumors. This shift may be driven by fumarate accumulation, which stabilizes HIF-1α, creating a pseudohypoxic environment that activates genes involved in cell proliferation, angiogenesis, and other hypoxia-adaptive pathways. In addition to HIF-1α stabilization, FH deficiency–induced fumarate accumulation activates several complementary mechanisms that further promote angiogenesis. For example, fumarate-mediated KEAP1 succination activates NRF2 signaling, indirectly supporting angiogenesis. Mitochondrial dysfunction in FH-deficient cells results in excess reactive oxygen species (ROS), which reinforces HIF-1α stabilization and VEGF induction. Moreover, fumarate has emerged as an oncometabolite that inhibits α-ketoglutarate (α-KG)–dependent dioxygenases, resulting in epigenetic alterations such as histone and DNA hypermethylation. These changes have been shown to drive epithelial-to-mesenchymal transition (EMT) [63,64,65,66,67]. Concurrently, the upregulation of angiogenesis, VEGF signaling, and epithelial-to-mesenchymal transition (EMT) suggests a tumor microenvironment favoring invasion and metastasis, similar to observations in HLRCC [5]. These findings reinforce the role of *FH* deficiency in enhancing angiogenesis through VEGF signaling and altering cellular metabolism.

In addition, our immune deconvolution analysis revealed an elevation of activated mast cells in *FH*-deficient tumors. Mast cells, traditionally associated with allergic responses, have emerged in recent years as important modulators of the TME. Tumor-associated mast cells can promote angiogenesis, invasion, and immune suppression through the release of pro-angiogenic mediators [61]. Although their role in BC remains complex and context-dependent, our findings support the possibility that mast cell elevation in *FH*-deficient tumors contributes to the observed pro-angiogenic phenotype.

The prognostic effect of *FH* deficiency in TCGA samples appeared to be subtype-specific, with a marked adverse impact on OS in basal tumors. This pattern suggests that *FH* deficiency may have particular importance in basal BC biology. In line with this, the presented case study involved a patient with TNBC, further supporting the relevance of *FH* deficiency in basal-like breast cancer.

To further validate these insights, we conducted weighted gene co-expression network analysis (WGCNA), identifying modules and hub genes strongly associated with *FH* expression. Key pathways enriched in the *FH*-deficient group included blood vessel development, cell adhesion, and VEGF signaling, underscoring the role of angiogenesis and metastasis in these tumors. Several hub genes, such as CDH5, CLDN5, VWF, and PECAM1, were significantly upregulated, reflecting their established involvement in vascular development and tumor progression [68,69,70,71,72,73,74].

The phenotypic similarity between BC tumors with *FH* deficiency and HLRCC tumors further highlights the therapeutic potential of targeting VEGF signaling in this subgroup. As a clinical example of these findings, we described a case of a BC patient with an *FH* mutation successfully treated with a VEGF inhibitor, a therapy commonly used for HLRCC. However, despite these promising findings, our study has certain limitations. The rarity of *FH* aberrations constrained both the size of the affected patient cohort and the availability of independent datasets for external validation, resulting in reliance on retrospective, publicly available resources. In this context, the case report presented here provides a form of clinical validation, though further prospective validation is warranted. Functional validation of these pathways in experimental models, including in vitro assays, as well as prospective clinical studies, is necessary to confirm the therapeutic relevance of *FH* deficiency. In addition, both the low prevalence of *FH* alterations and the possible contribution of endothelial content to RNA expression data limit the robustness of our findings, though they highlight the need for biomarker-based patient selection in future anti-VEGF trials. Furthermore, our angiogenesis findings are based on a curated MSigDB signature, and validation across independent cohorts and breast cancer subtypes is still required. Finally, the interplay between *FH*-driven metabolic changes and the tumor immune microenvironment remains an important area for future investigation.

## 5. Conclusions

Our findings suggest that *FH* aberrations, though rare, define a distinct molecular subset of BC characterized by metabolic reprogramming, enhanced angiogenesis, and upregulated VEGF signaling. This molecular profile supports the rationale for exploring VEGF inhibitors as a potential targeted therapy for *FH*-deficient BC. The identification of *FH* expression as a potential biomarker could help personalize anti-angiogenic treatment strategies in breast cancer patients, particularly those who currently lack effective targeted options.

## Figures and Tables

**Figure 1 cancers-17-02942-f001:**
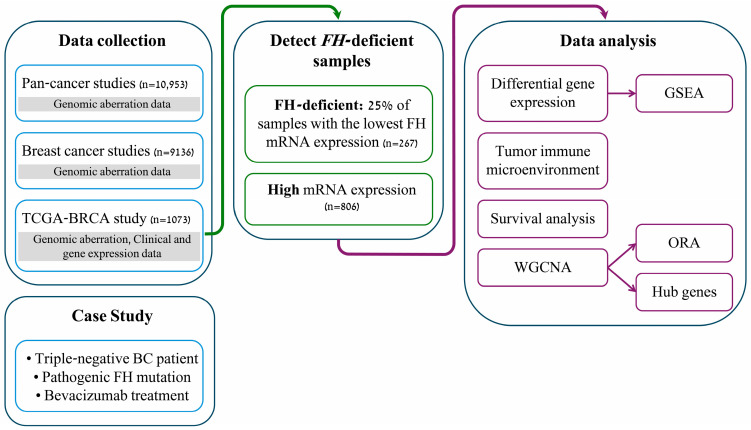
Study design and datasets.

**Figure 2 cancers-17-02942-f002:**
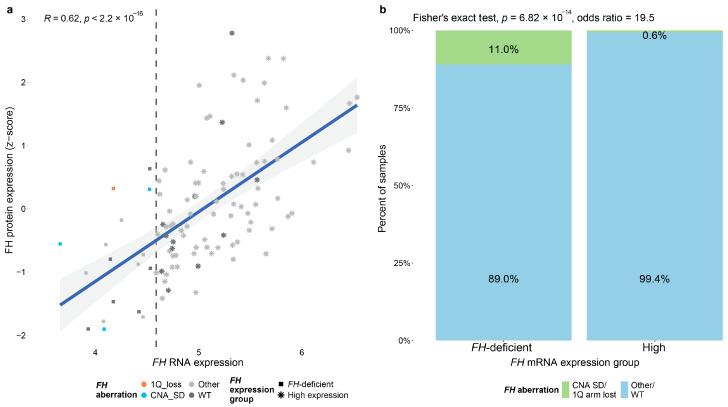
**FH expression in primary breast cancer tumors**. (**a**) Scatter plot representing the correlation between *FH* mRNA expression and FH protein expression from 105 samples. The color of the points represents types of FH aberration (orange—1Q arm lost; light blue—FH CNA shallow deletion (SD); light gray—other FH aberrations (1Q arm gain/FH CNA gain/amplification); dark gray—WT FH). Point shape represents the FH mRNA expression group (square—FH-deficient; star—high expression). The dashed line indicates the cut-off between *FH* expression groups. (**b**) Bar graph illustrating the percentage of samples with *FH* CNA SD or 1Q arm lost (green) in each of the *FH* mRNA expression groups.

**Figure 3 cancers-17-02942-f003:**
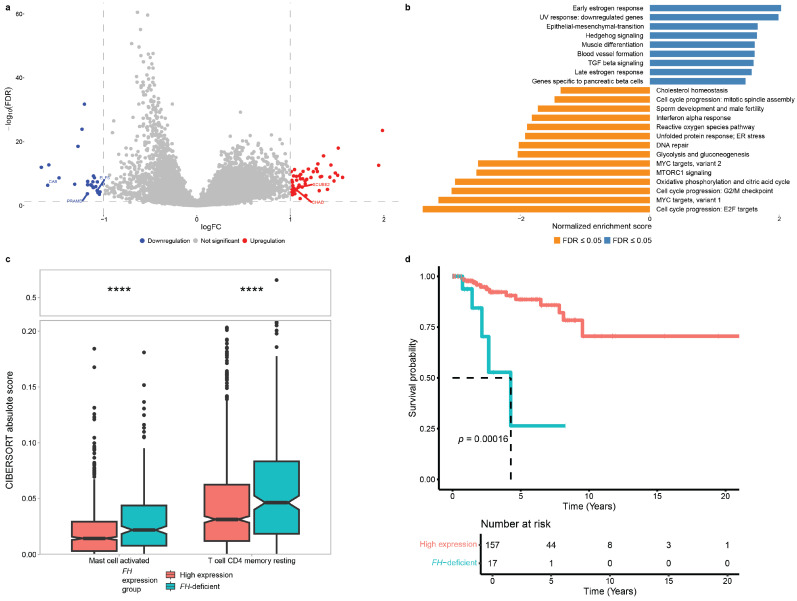
**Transcriptomic differences based on *FH* expression**. (**a**) Volcano plot of differentially expressed genes (DEGs) between *FH*-deficient and *FH* high-expression groups, showing log fold change (logFC) on the x-axis and -log10(FDR) on the y-axis. Point color indicates upregulated (red) or downregulated (blue) genes. Dashed lines mark statistical thresholds: horizontal line for FDR ≤ 0.05 and vertical lines for log fold-change > ±1 (**b**) Gene set enrichment analysis (GSEA) of DEGs comparing *FH*-deficient and *FH* high-expression groups. (**c**) Boxplots showing CIBERSORT absolute scores for immune cell subsets in TCGA breast cancer samples, stratified by *FH* expression (high vs. *FH*-deficient) (Wilcoxon test, ****: fdr < 0.0001). (**d**) Kaplan–Meier overall survival analysis of basal BC patients stratified by *FH* expression (red—high expression group; turquoise—*FH*-deficient group).

**Figure 4 cancers-17-02942-f004:**
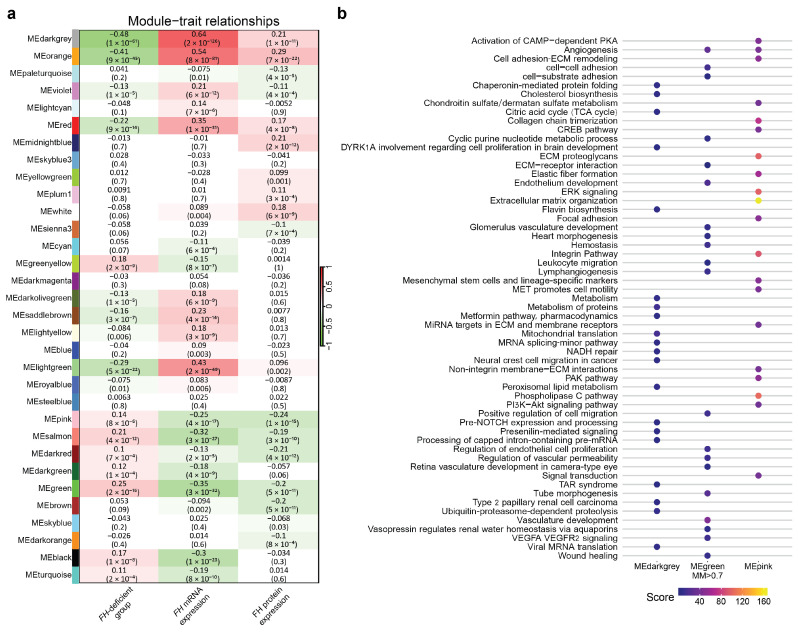
**WGCNA of *FH* expression**. (**a**) Module–trait relationships calculated by the WGCNA algorithm. Each cell shows the correlation coefficient between a gene module and a specific trait with the corresponding *p*-value in parentheses. (**b**) Pathway enrichment analysis of genes included in the MEdarkgrey, MEgreen (genes with ModelMembership (MM) > 0.7), and MEpink modules. The point size represents the enrichment score (−log2(FDR)).

**Figure 5 cancers-17-02942-f005:**
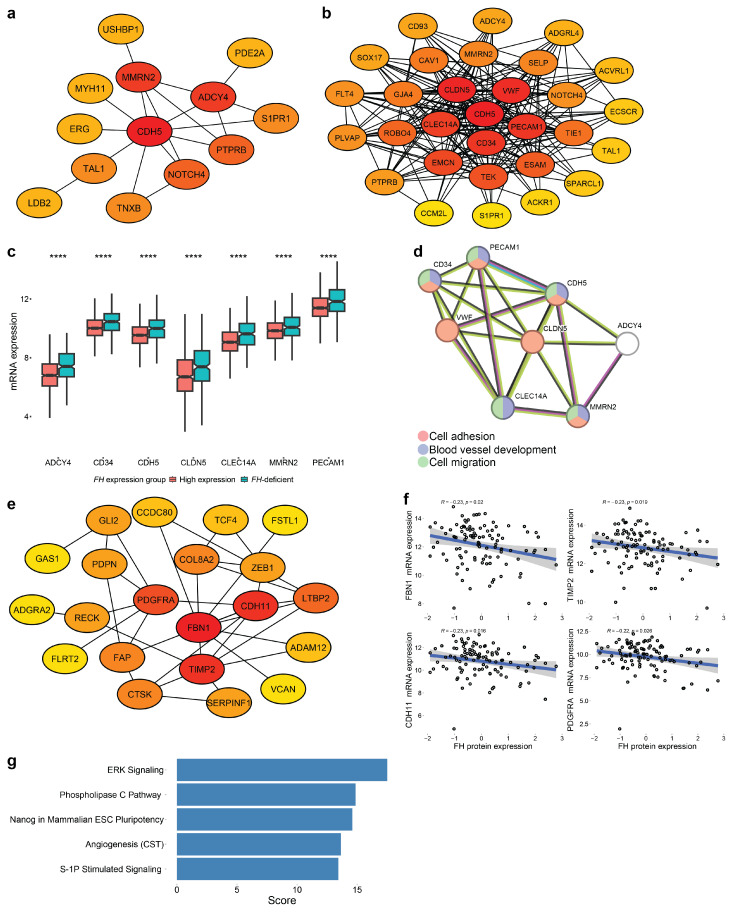
Hub genes in WGCNA modules. (**a**,**b**) Protein–protein interaction (PPI) network of hub genes detected in the MEgreen module according to (**a**) *FH*-deficient expression group (yes/no) and (**b**) *FH* mRNA expression (continuous values), visualized by the Cytoscape software (Version 3.10.2). Colored circles represent the connectivity degree of the hub gene. Red nodes correspond to higher degrees of connectivity. (**c**) Boxplot of mRNA expression of the seven key hub genes in the MEgreen module according to *FH* expression group (****: *p*-value ≤ 0.0001). (**d**) PPI network of the eight key hub genes in the MEgreen module visualized by STRING-DB software (Version 12.0). (**e**) PPI network of hub genes of the MEpink module according to FH protein expression, visualized by the Cytoscape software. Colored circles represent the connectivity degree of the hub gene. Red nodes correspond to higher degrees of connectivity. (**f**) Scatter plots that represent the correlation between FH protein expression (x-axis) and the expression of the four key hub genes (y-axis) in the MEpink module. For each scatter plot, Spearman correlation coefficients (R) and the corresponding *p*-values are presented. (**g**) The top five enriched pathways of the MEpink module key hub genes. The score represents −log2(FDR) of the enrichment.

**Figure 6 cancers-17-02942-f006:**
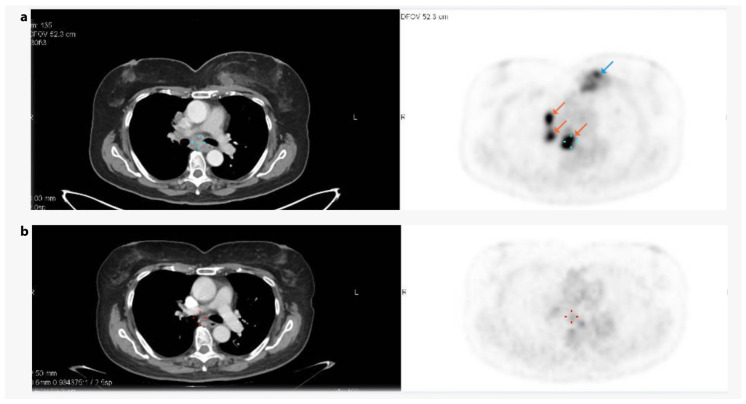
**CT and PET imaging of the patient with *FH* mutation**. (**a**) Before treatment with paclitaxel (80 mg/m^2^ weekly) and bevacizumab (10 mg/kg every 2 weeks), and (**b**) after three months of treatment. The primary breast mass is highlighted using a blue arrow, while the metastases are highlighted in orange.

**Table 1 cancers-17-02942-t001:** *FH* aberrations in breast cancer studies.

Study	Sample Type	Total (N)	*FH* Mutation		*FH* Structural Variance (SV)		*FH* Copy Number Alterations (CNAs)						1Q Arm Status		
			Number of Mutated Samples	Total Number of Samples	Number of Samples with SV	Total Number of Samples	Amp. *	Gain	SD *	DD *	Diploid	Total Number of Samples	Gain	Loss	Total Number of Samples
Breast Cancer (MSK, *Cancer Cell* 2018 [17])	Metastasis	1000	2 (0.2%)	1000	1 (0.1%)	1000	20	0	0	1 (0.1%)	979	1000	NA	NA	NA
	Primary	918	4 (0.44%)	918	1 (0.11%)	918	1	0	0	0	917	918			
Breast Cancer (MSK, *Nature Cancer* 2020) [18]	Metastasis	30	0	30	0	30	1	1	0	0	28	30	NA	NA	NA
	Primary	8	0	8	0	8	0	0	0	0	8	8			
MAPK on Resistance to Anti-HER2 Therapy for Breast Cancer (MSK, *Nat Commun*. 2022) [19]	Metastasis	91	0	91	0	91	1	0	0	0	90	91	NA	NA	NA
	Primary	54	0	54	0	54	0	0	0	0	54	54			
Metastatic Breast Cancer (MSK, *Cancer Discovery* 2022) [20]	Metastasis	1365	6 (0.44%)	1365	1 (0.07%)	1365	25	0	0	0	1340	1365	NA	NA	NA
MSK MetTropism (MSK, *Cell* 2021) [21]	Metastasis	1048	4 (0.38%)	1048	1 (0.1%)	1048	19	0	0	1 (0.1%)	1028	1048	NA	NA	NA
	Primary	1561	6 (0.38%)	1561	1 (0.06%)	1561	6	0	0	0	1555	1561			
MSK-IMPACT Clinical Sequencing Cohort (MSK, *Nat Med* 2017) [22]	Metastasis	837	1 (0.12%)	837	1 (0.12%)	837	14	0	0	1 (0.12%)	822	837	NA	NA	NA
	Primary	500	2 (0.4%)	500	1 (0.2%)	500	1	0	0	0	499	500			
Non-CDH1 Invasive Lobular Carcinoma (MSK, 2023) [23]	Primary	25	0	25	0	25	0	0	0	0	25	25	NA	NA	NA
China Pan-cancer (OrigiMed, *Nature* 2022) [24]	Metastasis	25	0	25	0	25	0	0	0	0	25	25	NA	NA	NA
	Primary	71	0	71	0	71	0	0	0	0	71	71			
Breast Invasive Carcinoma (TCGA, PanCancer Atlas) [25]	Primary	1052	4 (0.38%)	1052	0	1052	101	667	37 (3.5%)	0	247	1052	625	10 (1.13%)	888
The Metastatic Breast Cancer Project (provisional, December 2021) [26]	NA	334	1 (0.3%)	334	0	156	49	115	96 (28.7%)	6 (1.8%)	68	334	NA	NA	NA
MSK-CHORD (MSK, *Nature* 2024) [27]	Metastasis	2564	9 (0.35%)	2564	1 (0.04%)	2464	42	0	0	1 (0.04%)	2421	2464	NA	NA	NA
	Primary	2859	13 (0.45%)	2859	1 (0.034%)	2859	14	0	0	0	2845	2859			
Breast Cancer (MSK, 2025) [28]	Metastasis	2048	12 (0.58%)	2048	2 (0.1%)	2048	38	0	0	1 (0.05%)	2009	2048	NA	NA	NA
	Primary	1812	5 (0.27%)	1812	1 (0.06)	1812	8	0	0	0	1804	1812			

* Amp—amplification; SD—shallow deletion; DD—deep deletion.

**Table 2 cancers-17-02942-t002:** Characterization of the mutational landscape of likely oncogenic *FH* mutations in breast cancer tumors.

Patient ID	Sample ID	Study	Sample Type	TMB	Protein Change	Mutation Type	Variant Type	Mutation Status	Chr.	Start Pos	End Pos	Ref	Var	Fraction Genome Altered
P-0022525	P-0022525-T01-IM6	MSK, Cell 2021 [21]	P	3.46	L208Vfs*9	Frameshift Insertion	INS	Somatic	1	241,672,019	241,672,020	-	C	0.222
P-0005712	P-0005712-T01-IM5	MSK, Cell 2021 [21]	P	3.91	*FH* intragenic	Fusion	Dup	Somatic	1	241,665,729	NA			0.4247
P-0005712	P-0005712-T01-IM5	MSK, Cancer Cell 2018 [17]	P	0.133	*FH* intragenic	Fusion	Dup	Somatic	1	241,665,729	NA			0.5076
P-0045182	P-0045182-T01-IM6	MSK, Nature 2024 [27]	P	1.73	Q237*	Nonsense	SNP	Somatic	1	241,671,932	241,671,932	G	A	0.4
Patient0707	P-0707	OrigiMed, Nature 2022 [24]	P	0.166	X186_splice	Splice Region	SNP	NA	1	241,672,089	241,672,089	T	C	NA
P-0017116	P-0017116-T01-IM6	MSK, Cancer Discovery 2022 [20]	M	8.65	P63Ifs*9	Frameshift Deletion	DEL	Somatic	1	241,680,556	241,680,562	CATTTGG	-	0.2691
P-0004918	P-0004918-T02-IM6	MSK, Cell 2021 [21]	M	6.92	*RGS7-FH* Fusion	Fusion	Dup	Somatic	1	241,357,653	NA			0.2868
P-0000532	P-0000532-T02-IM5	MSK, Cancer Discovery 2022 [20]	M	8.81	*PDE1C-FH* Fusion	Fusion	Trans	Somatic	7	31,926,696	NA			0.6164
P-0048392	P-0048392-T01-IM6	MSK, Nature 2024 [27]	M	8.65	MIR-1273E/1273E-*FH* Fusion	Fusion	Inversion	Somatic	1	240,716,629	NA			0.136
P-0000532	P-0000532-T02-IM5	MSK, Cancer Cell 2018 [17]	M	0.3	*PDE1C-FH* Fusion	Fusion	Trans	Somatic	7	31,926,696	NA			0.5412
P-0008574	P-0008574-T03-IM6	MSK, Cell 2021 [21]	M	7.78	M336K	Missense	SNP	Somatic	1	241,667,443	241,667,443	A	T	0.2097

P—primary; M—metastasis; Dup—duplication; Trans—translocation.

## Data Availability

The TCGA dataset used in this study is publicly available through the metaGXbreast R package [31] (https://bioconductor.org/packages/release/data/experiment/html/MetaGxBreast.html, accessed on 1 January 2022) and cBioPorta [15,16] (https://www.cbioportal.org/study/summary?id=brca_tcga_pan_can_atlas_2018, accessed on 17 April 2024). Other pan-cancer and breast cancer datasets are publicly available through the cBioPortal. On request, the analyzed data generated during this study can be provided by the corresponding authors.

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
