# Peer review of "Revealing the Angiogenic Signature of FH-Deficient Breast Cancer: Genomic Profiling and Clinical Implications"

_cancers, 2025, doi:10.3390/cancers17182942_

Round 1

Reviewer 1 Report

Comments and Suggestions for Authors

The study found that FH deficiency promotes a tumor microenvironment conducive to angiogenesis and may serve as a predictive biomarker for VEGF-targeted therapy. The article flows logically from big data to multi-omics and then to real-world applications, demonstrating clinical value, but I have the following suggestions:

  1. FH-deficient breast cancer was only 0.36% among 9,163 cases, with too few cases. It is suggested to provide a statistical power calculation or discuss the risk of false negatives.

  1. Age, tumor size, and molecular subtype may all simultaneously affect FH expression and angiogenic signaling. It is suggested that a multivariate regression model (logistic or Cox) could be supplemented to verify whether FH deficiency is an independent factor.

  1. Metabolic reprogramming often interacts with the immune microenvironment. It is suggested that the differences in immune cell abundance in the low FH group could be analyzed, and the potential synergistic effects of immunotherapy could be discussed.

  1. All data are from public databases and lack prospective cohorts. It is suggested that external validation can be added.

  1. It is suggested that a uniform standard be provided when "FH-deficient" first appears in the main text.

  1. The picture resolution, font, caption, etc. are not clear. It is recommended to use 600 dpi or above.

  1. It is suggested to refine the parameter selection of WGCNA to ensure the rationality of the soft threshold and module merging.

  1. It is suggested that the sample has passed batch correction.

  1. The article only infers the loss of FH function at the mRNA/protein/copy number level. It is suggested that in vitro functional experiments be conducted.

Author Response

We would like to thank the reviewer for his constructive feedback. We appreciate the time and effort spent reviewing our revised manuscript. The suggestions and comments have been very helpful in guiding our revisions, and we believe the manuscript has been significantly improved as a result. Detailed responses to each comment are provided below. We hope these changes adequately address the concerns raised and improve the manuscript's overall quality.

Comment 1:

FH-deficient breast cancer was only 0.36% among 9,163 cases, with too few cases. It is suggested to provide a statistical power calculation or discuss the risk of false negatives.

Response 1:

As suggested, we calculated the mean prevalence of FH mutations across the 13 breast cancer studies included in our analysis, which was 0.36% (95% CI: 0.29%–0.43%). This confidence interval demonstrates that, despite the rarity of events, the estimate is reasonably precise across studies. Nevertheless, we acknowledge that the low frequency reduces statistical power and increases the risk of false negatives, particularly in smaller cohorts. To address this, we revised the Results section (p.6, lines:174-178).

Comment 2:

Age, tumor size, and molecular subtype may all simultaneously affect FH expression and angiogenic signaling. It is suggested that a multivariate regression model (logistic or Cox) could be supplemented to verify whether FH deficiency is an independent factor.

Response 2:

As suggested, we performed univariate and multivariate linear regression analyses with angiogenesis score as the dependent variable and FH expression, age, tumor size, and PAM50 subtype as predictors (Supplementary Table 3), the results shows that FH expression remains a highly significant predictor of angiogenesis (p = 1.97 × 10⁻¹⁴), independent of age, tumor size, or molecular subtype. We have revised the Results section accordingly to reflect these findings (p.8, lines: 223-229).

Comment 3:

Metabolic reprogramming often interacts with the immune microenvironment. It is suggested that the differences in immune cell abundance in the low FH group could be analyzed, and the potential synergistic effects of immunotherapy could be discussed.

Response 3:

As suggested we examine immune infiltration in the TME, main finding showed the increase in both activated mast cells and resting memory CD4⁺ T cells (Figure 2C). We have updated the results and discussion sections, main findings regarding to mast cells elevation in the FH-deficiency tumors (p.10, lines:253-260, Figure 3C, and Supplementary Table 5, p.15, lines:372-378).

Comment 4:

All data are from public databases and lack prospective cohorts. It is suggested that external validation can be added.

Response 4:

We agree that external validation would strengthen the conclusions. Nevertheless, given the low prevalence of FH aberrations across breast cancer datasets, it was not feasible to identify an additional validation cohort. The presented case report serves as a unique external clinical validation in support of our bioinformatic analysis. This limitation has been acknowledged in the discussion section (p.15, lines:397-399)

Comment 5:

It is suggested that a uniform standard be provided when "FH-deficient" first appears in the main text.

Response 5:

The distinction between FH-deficient and FH-low expression may have been unclear. To provide a consistent standard, we now explicitly define FH-deficient in the Introduction as reduced or absent FH activity (p.2 lines:76) and replaced all instances of FH-low expression with FH-deficient throughout the manuscript.

Comment 6:

The picture resolution, font, caption, etc. are not clear. It is recommended to use 600 dpi or above.

Response 6:

All figures are now provided as separate high-quality files with a minimum resolution of 600 dpi to improve readability.

Comment 7:

It is suggested to refine the parameter selection of WGCNA to ensure the rationality of the soft threshold and module merging.

Response 7:

The WGCNA parameters were selected according to the default recommendations for a signed network, ensuring that the soft threshold reached a scale-free topology fit index above 0.8. Module merging was performed at a cut height of 0.25. The corresponding diagnostic plots have been added as Supplementary Figure 1. We also clarified this in the text (p.5, lines:135-138).

Comment 8:

It is suggested that the sample has passed batch correction.

Response 8:

Since all samples analyzed here belong to one dataset (TCGA-BRCA) within the MetaGxBreast collection, no cross-study variability exists. For this reason, batch correction was not performed, unlike in prior published studies (doi: 10.1038/s41598-023-48002-x, 10.1016/j.esmoop.2025.105080) where multiple datasets were combined.

Comment 9:

The article only infers the loss of FH function at the mRNA/protein/copy number level. It is suggested that in vitro functional experiments be conducted.

Response 9:

We recognize the importance of functional experiments to directly validate the biological consequences of FH loss. While such assays were beyond the scope of the current study, we provided clinical validation in the form of a case report of an FH-mutated BC patient who responded to VEGF inhibition. Accordingly, we have revised the limitations section to explicitly note the need for in-vitro validation of FH deficiency in BC (p.15, line:400).

Reviewer 2 Report

Comments and Suggestions for Authors

Dear Editor

Thank you for the opportunity to review this interesting manuscript entitled “Revealing the Angiogenic Signature of FH-Deficient Breast Cancer: Genomic Profiling and Clinical Implications.” The study is well executed, integrates genomic and transcriptomic profiling with clinical relevance, and has clear potential impact. I recommend accepting minor revisions.

Comments to the Authors:

  1. The English language should be improved throughout the manuscript to enhance grammar, sentence flow, and clarity.
  2. The Discussion section requires more elaboration. Please expand on how FH deficiency may promote angiogenesis. Possible mechanisms include HIF-1α stabilization leading to VEGF induction (pseudohypoxia), KEAP1–NRF2/redox signaling, mitochondrial dysfunction and ROS signaling, and epigenetic effects caused by fumarate accumulation.
  3. If available, please cite any studies showing FH knockdown or deficiency promoting angiogenesis in vivo animal models. If no such studies exist, kindly acknowledge this as a limitation and suggest it as a future research direction.
  4. The figures would benefit from improved quality and presentation.
  5. Please add a short note in the Discussion about the prevalence of FH alterations in your cohort and mention the possible confounding effect of endothelial content in bulk RNA-seq. Also highlight how these findings may guide biomarker-driven patient selection for anti-VEGF therapy.

This is a valuable study. Addressing these comments will further strengthen it.

Comments on the Quality of English Language

Need to be Improved

Author Response

# Reviewer 2

We would like to thank the reviewer for his constructive feedback. We appreciate the time and effort spent reviewing our revised manuscript. The suggestions and comments have been very helpful in guiding our revisions, and we believe the manuscript has been significantly improved as a result. Detailed responses to each comment are provided below. We hope these changes adequately address the concerns raised and improve the manuscript's overall quality.

Comment 1:

The English language should be improved throughout the manuscript to enhance grammar, sentence flow, and clarity.

Response 1:

The manuscript has undergone professional English editing by the university’s academic editor, and the official confirmation letter is attached.

Comment 2:

The Discussion section requires more elaboration. Please expand on how FH deficiency may promote angiogenesis. Possible mechanisms include HIF-1α stabilization leading to VEGF induction (pseudohypoxia), KEAP1–NRF2/redox signaling, mitochondrial dysfunction and ROS signaling, and epigenetic effects caused by fumarate accumulation.

Response 2:

We have expanded the Discussion to include a mechanistic overview of how FH deficiency can promote angiogenesis (p.14, lines:357-367).

Comment 3:

If available, please cite any studies showing FH knockdown or deficiency promoting angiogenesis in vivo animal models. If no such studies exist, kindly acknowledge this as a limitation and suggest it as a future research direction.

Response 3:

The revised Introduction now cites in vivo experiments using conditional Fh1 knockout mice. Although these studies focused on cyst development, we point out that the reported HIF and VEGF pathway activation provides relevant evidence for the angiogenic consequences of FH deficiency (p.2, lines:84-86).

Comment 4:

The figures would benefit from improved quality and presentation.

Response 4:

All figures are now provided as separate high-quality files with a minimum resolution of 600 dpi to improve readability.

Comment 5:

Please add a short note in the Discussion about the prevalence of FH alterations in your cohort and mention the possible confounding effect of endothelial content in bulk RNA-seq. Also highlight how these findings may guide biomarker-driven patient selection for anti-VEGF therapy.

Response 5:

We expanded the limitations section in the discussion to mention the rarity of FH alterations, possible bias from endothelial content in RNA-seq data, and the importance of these findings for biomarker-based use of anti-VEGF therapy (p.15, lines:402-405).

Reviewer 3 Report

Comments and Suggestions for Authors

This manuscript seeks to uncover the angiogenesis-related molecular landscape in FH-deficient breast cancer, using multi-omics bioinformatics profiling with clinical correlation. The scientific topic is meaningful and may provide potential implications for individualized anti-angiogenic therapy. The general writing is fluent and most of the figures are well-prepared. However, the current version still contains several issues in methodology, presentation logic, and data interpretation.
Comment 1: The manuscript described that FH deficiency might induce a pro-angiogenic phenotype in breast cancer based on transcriptomic analysis and pathway enrichment results. However, all findings are currently correlative in nature. Without providing direct functional data, it is difficult to conclude the biological causality between FH downregulation and angiogenesis activation. Therefore, I would like to kindly suggest that the authors consider including more validation data. These additions will greatly strengthen the scientific foundation of the mechanistic insight.
Comment 2: In the current study, a specific gene set was used to define the angiogenic profile of FH-deficient cases, yet the construction pipeline is not clearly explained. It remains unclear whether this gene set was obtained from public angiogenesis-related databases (such as MSigDB or Gene Ontology) or through de novo differential expression analysis. In addition, the reproducibility of this gene signature in external datasets or across breast cancer subtypes is not yet discussed. I would recommend the authors to further explain the derivation method of the angiogenic signature and consider performing external dataset validation to confirm its stability and specificity.
Comment 3: Although the authors have provided data regarding FH expression and gene signature levels in breast cancer patients, the clinical impact remains underexplored. For instance, the manuscript does not examine whether the FH-low or angiogenic-high group has worse clinical outcomes such as overall survival or relapse-free survival. Also, breast cancer is a heterogeneous disease composed of multiple molecular subtypes. Without proper stratification (e.g., luminal A/B, HER2+, TNBC), the interpretation might be confounded. It is suggested to conduct Kaplan–Meier survival analysis and multivariable Cox regression, along with subtype-based analysis, to better illustrate the clinical relevance of the findings.
Comment 4: The layout and arrangement of figures are fragmented and lack coherence. It is kindly noted that the figures are presented in a relatively scattered manner. For example, the volcano plots, heatmaps, and pathway enrichment results are separated into different figure blocks, making it difficult for the reader to understand the flow of scientific logic. It would be more reader-friendly to organize them into integrated panels, such as combining DEG analysis and pathway analysis into one main figure. 
Comment 5: The manuscript would benefit from a comprehensive workflow or schematic representation of the research methodology. This would help readers follow the experimental design and analytical processes more easily. I suggest that the authors consult recent research articles in MDPI as exemplary models for structuring this aspect of their study (PMID: 35328243, 34834441, 30669548). Meanwhile, most Figures are too low in resolution; please provide high-resolution versions with clearer legends. Most figures have very small font sizes and overlapping annotations. Uniform color usage and enlarging key labels would greatly help readers.

Author Response

We would like to thank the reviewer for his constructive feedback. We appreciate the time and effort spent reviewing our revised manuscript. The suggestions and comments have been very helpful in guiding our revisions, and we believe the manuscript has been significantly improved as a result. Detailed responses to each comment are provided below. We hope these changes adequately address the concerns raised and improve the manuscript's overall quality.

Comment 1:

The manuscript described that FH deficiency might induce a pro-angiogenic phenotype in breast cancer based on transcriptomic analysis and pathway enrichment results. However, all findings are currently correlative in nature. Without providing direct functional data, it is difficult to conclude the biological causality between FH downregulation and angiogenesis activation. Therefore, I would like to kindly suggest that the authors consider including more validation data. These additions will greatly strengthen the scientific foundation of the mechanistic insight.

Response 1:

We agree that external validation would strengthen the conclusions. Nevertheless, given the low prevalence of FH aberrations across breast cancer datasets, it was not feasible to identify an additional validation cohort. The presented case report serves as a unique external clinical validation in support of our bioinformatic analysis. This limitation has been acknowledged in the discussion section (p.15, lines:397-399)

Comment 2:

In the current study, a specific gene set was used to define the angiogenic profile of FH-deficient cases, yet the construction pipeline is not clearly explained. It remains unclear whether this gene set was obtained from public angiogenesis-related databases (such as MSigDB or Gene Ontology) or through de novo differential expression analysis. In addition, the reproducibility of this gene signature in external datasets or across breast cancer subtypes is not yet discussed. I would recommend the authors to further explain the derivation method of the angiogenic signature and consider performing external dataset validation to confirm its stability and specificity.

Response 2:

We clarified that the “angiogenic signature” in our study was based on the upregulation of the HALLMARK_ANGIOGENESIS pathway from the MSigDB Hallmark collection, supported by the expression of individual angiogenesis-related genes previously reported in the literature. We have revised the Methods and Results sections to indicate the source of the pathway (p.6, lines:153-154; p.9, line:248). We also added a sentence in the discussion regarding the limitations noting that external validation of this angiogenic signature remains to be conducted (p.15, lines:405-407).

Comment 3:

Although the authors have provided data regarding FH expression and gene signature levels in breast cancer patients, the clinical impact remains underexplored. For instance, the manuscript does not examine whether the FH-low or angiogenic-high group has worse clinical outcomes such as overall survival or relapse-free survival. Also, breast cancer is a heterogeneous disease composed of multiple molecular subtypes. Without proper stratification (e.g., luminal A/B, HER2+, TNBC), the interpretation might be confounded. It is suggested to conduct Kaplan–Meier survival analysis and multivariable Cox regression, along with subtype-based analysis, to better illustrate the clinical relevance of the findings.

Response 3:

We thank the reviewer for this highly important comment. As suggested, we performed Kaplan–Meier survival and recurrence analyses stratified by PAM50 subtypes and FH expression groups (Supplementary Table S6), FH-deficient tumors were significantly associated with worse overall survival in the basal subtype (HR = 6.0, 95% CI 2.09–17.23, p = 0.000875), while no significant associations were observed in the other subtypes. Disease-free survival did not show consistent differences across subtypes. These results suggest that the adverse prognostic effect of FH deficiency may be subtype-specific, with particular impact in basal BC, indeed, the patient we described in the study case was with TNBC. We incorporate these findings in results and discussion sections (p.10, lines:261-267; p.15, lines:379-383).

Comment 4:

The layout and arrangement of figures are fragmented and lack coherence. It is kindly noted that the figures are presented in a relatively scattered manner. For example, the volcano plots, heatmaps, and pathway enrichment results are separated into different figure blocks, making it difficult for the reader to understand the flow of scientific logic. It would be more reader-friendly to organize them into integrated panels, such as combining DEG analysis and pathway analysis into one main figure.

Response 4:

We recognize the reviewer’s concern regarding figure arrangement. The source of this confusion is due to two distinct pathway enrichment approaches in our study: GSEA, which was performed directly on the DEG results, and ORA, which was conducted on genes out of selected WGCNA modules. Because these analyses rely on different input gene sets and serve complementary purposes, we chose to present them separately to maintain methodological clarity. To improve clarity, we revised the Methods and Results to better highlight this distinction (p.6, lines:154, 1157; p.9, line:248; p.10, lines:277-280).

Comment 5:

The manuscript would benefit from a comprehensive workflow or schematic representation of the research methodology. This would help readers follow the experimental design and analytical processes more easily. I suggest that the authors consult recent research articles in MDPI as exemplary models for structuring this aspect of their study (PMID: 35328243, 34834441, 30669548). Meanwhile, most Figures are too low in resolution; please provide high-resolution versions with clearer legends. Most figures have very small font sizes and overlapping annotations. Uniform color usage and enlarging key labels would greatly help readers.

Response 5:

We have incorporated a schematic workflow of the study design in the Methods section (new Figure 1) to provide a clear overview of the research process. In addition, all figures were replaced with high-resolution versions.